# Endemic Melioidosis in Southern China: Past and Present

**DOI:** 10.3390/tropicalmed4010039

**Published:** 2019-02-25

**Authors:** Xiao Zheng, Qianfeng Xia, Lianxu Xia, Wei Li

**Affiliations:** 1State Key Laboratory for Infectious Disease Prevention and Control, Collaborative Innovation Center for Diagnosis and Treatment of Infectious Diseases, National Institute for Communicable Disease Control and Prevention, Chinese Center for Disease Control and Prevention, Beijing 102206, China; xialianxu@icdc.cn (L.X.); liwei@icdc.cn (W.L.); 2Chinese Field Epidemiology Training Program, Chinese Center for Disease Control and Prevention, Beijing 102206, China; 3Laboratory of Tropical Biomedicine and Biotechnology, School of Tropical Medicine and Laboratory Medicine, Hainan Medical University, Haikou 571199, China; xiaqianfeng@sina.com

**Keywords:** melioidosis, *Burkholderia pseudomallei*, epidemiology, China

## Abstract

Melioidosis is a severe tropical infectious disease caused by the soil-dwelling bacterium *Burkholderia pseudomallei*, predominantly endemic to Southeast Asia and northern Australia. Between the 1970s and the 1990s, the presence of *B. pseudomallei* causing melioidosis in humans and other animals was demonstrated in four coastal provinces in southern China: Hainan, Guangdong, Guangxi, and Fujian, although indigenous cases were rare and the disease failed to raise concern amongst local and national health authorities. In recent years, there has been a rise in the number of melioidosis cases witnessed in the region, particularly in Hainan. Meanwhile, although China has established and maintained an effective communicable disease surveillance system, it has not yet been utilized for melioidosis. Thus, the overall incidence, social burden and epidemiological features of the disease in China remain unclear. In this context, we present a comprehensive overview of both historical and current information on melioidosis in Southern China, highlighting the re-emergence of the disease in Hainan. Surveillance and management strategies for melioidosis should be promoted in mainland China, and more research should be conducted to provide further insights into the present situation.

## 1. Introduction

Melioidosis is a fatal infectious disease caused by *Burkholderia pseudomallei*, a saprophytic environmental bacterium that is endemic in many tropical regions of the world and affects both humans and animals [1,2]. In humans, although most organs of the body can be infected, acute pneumonia and septicaemia often represent the most common clinical manifestations, and are associated with a high mortality rate (10%–30%) for the disease [1,2]. Given the increase in the number of reported cases, as well as the extension of endemic regions in past decades, melioidosis is deemed to be a re-emerging infectious disease in many tropical countries [3], particularly in Southeast Asia and north Australia. More strikingly, a recent modeling study predicted that the true global burden of melioidosis was approximately 165,000 cases and 89,000 deaths a year [4], much higher than the total number of cases that are documented or reported. Although the true disease burden remains difficult to determine, it appears likely that there has been a dramatic underestimation of its scale in part due to misdiagnosis and underreporting in vast areas of the world appropriate for its endemicity. It is thus important to raise the profile of the disease across the tropical world.

Similar challenges have been encountered in mainland China. The environmental presence of *B. pseudomallei* was initially demonstrated in the 1970s and the first human case of melioidosis was identified in 1989 [5]; nonetheless, as only a few cases were identified at the time, attention soon declined and the disease was neglected for many years. However, with the rapid economic development and social transition over the past two decades, the situation has changed and there has been a substantial increase in the number of cases diagnosed in Hainan, with the occurrence of sporadic locally-acquired or imported cases in other areas throughout China. Clearly this shift requires a reconsideration of the current status and epidemiology of the disease within China. However, thanks to both preconceptions about the insignificance of the disease, and the fact that it has not been listed as a nationally notifiable infectious disease, studies of the epidemiology of melioidosis in China are few and far between and it is difficult to obtain resources to study it further.

Here, we provide an overview of the historical and current aspects of melioidosis in China by reviewing available data and information about human and animal melioidosis and *B. pseudomallei* in mainland China, with an emphasis on its resurgence in Hainan, including some publications that are only available in Chinese. We have excluded Hong Kong and Taiwan, which are both covered elsewhere in this issue. We hope that this review will help to inform clinicians and policy makers about the disease in southern China and lead to the formulation of plans to prevent and control it in the future.

## 2. History

The first event linking melioidosis to China occurred in France in the early 1970s when an outbreak of melioidosis at the Jardin des Plantes zoo was attributed to a giant panda from China [6]. Nevertheless, as pandas uniquely inhabit the high-altitude mountainous areas of Southwest China, where neither *B. pseudomallei* nor human or animal melioidosis has ever been reported, the panda might actually have been a victim of this outbreak rather than the source.

In mainland China, the discovery of environmental *B. pseudomallei* preceded the identification of the first human case of melioidosis by over 10 years. Since the early 1970s, horse farms and military establishments in Hainan and Guangdong had reported an increase of mallein test positive horses without any clinical features suggestive of glanders. A comprehensive environmental survey in the vicinity of mallein-positive stables was undertaken to investigate whether *B. pseudomallei* might be a potential cause of this phenomenon [7]. Between April and June 1975, Li and colleagues obtained 23 isolates of *B. pseudomallei* from 4.9% of pond waters sampled near these farms by inoculation in golden hamsters. Subsequent experiments showed that horses infected with *B. pseudomallei* gave positive mallein reactions [8]. From 1976 to 1979, the survey was extended to other southern provinces and another nine environmental isolates were obtained from Guangdong and Guangxi, two subtropical provinces close to Hainan, with no *B. pseudomallei* being cultured further north than this [8]. Subsequently, *B. pseudomallei* was isolated in 1985 from a paddy field in Putian, Fujian province, a subtropical region adjacent to Taiwan [9]. The first cases of culture-positive melioidosis were detected in 1982, when Lu isolated *B. pseudomallei* from slaughtered pigs in Hainan [10]. In 1981, Li found a prevalence of seropositivity (indirect hemagglutination (IHA) titer >1:40) ranging from 6.0% to 13.7% among people inhabiting the coastal areas of Hainan [8]. It thus appeared that exposure to *B. pseudomallei*, and therefore human melioidosis, was also likely to be present in this region.

Efforts to find cases of human melioidosis in Hainan continued throughout the 1980s, but most attempts were in vain. Eventually, the first human case of culture-positive melioidosis, presenting with an ulcer on his left leg, was identified in Sanya City in 1989 [11]. Two patients with fatal septicaemic melioidosis, both local farmers, were reported in 1990 from Zhanjiang City in Guangdong, which is located at the southernmost peninsula of mainland China opposite Hainan island [11]. Subsequently, Song et al. conducted a prospective survey looking for human cases at the largest provincial hospital on Hainan, Hainan People’s Hospital, in 1995 [12]. Within a one-year period, a total of eight culture-confirmed cases were identified, four of whom had acute septicaemia and the others chronic forms with abscesses in different organs. As elsewhere, seasonal and occupational associations were seen, as all cases were farmers and most presented during the rainy season in Hainan (May to October). Based on these findings, it was concluded that animal and human melioidosis were rare in the region and few further studies were conducted on it during the 1990s.

## 3. Review of Melioidosis Cases and Resurgence of the Disease in Hainan

During the 21st century, particularly after the severe acute respiratory syndrome (SARS) outbreak in 2003, China’s infectious disease surveillance, prevention and control systems expanded and improved considerably. Since then, the overall incidence of notifiable infectious diseases has been declining and some are even close to eradication [13]. By contrast, melioidosis has been one of the few infectious diseases whose incidence has increased in recent years, especially in the Hainan province. Clinical staff began to recognize increasing numbers of melioidosis cases in general hospitals in Hainan and their awareness of the infection grew [14,15]. For example, compared with a total of eight cases in 1996, approximately 20–30 cases were seen each year at Hainan People’s Hospital during the 2010s [15,16]. As culture facilities and diagnostic guidelines were already well established in this institution, it is likely that this reflected a genuine increase in incidence.

This increased incidence drew the attention of local and national health authorities and so in 2011 the Institute for Communicable Disease Control and Prevention of China (ICDC) established a working group with the aim of identifying and monitoring the incidence and epidemiology of melioidosis throughout China. Following this, a preliminary sentinel network representing the major government and teaching hospitals within Hainan was established, resulting in the identification of a total of 396 culture-confirmed cases (392 from Hainan, three from Guangxi and one from Guangdong) between 2002 and 2016 (Figure 1). The available medical records of 289 cases (all from Hainan) were reviewed to establish information about the demographics, clinical features and outcomes. It was found that the patients were distributed around the periphery of Hainan Island with no cases originating from the two central mountainous prefectures (Wuzhishan and Qiong Zhong) as seen in Figure 2; three major port cities (Sanya, Haikou, and Dongfang) contributed to nearly half the cases (123, 42.6%). Of 289 patients for whom data were available, 245 were male and 44 were female; apart from one neonate (17 days), the other patients ranged from one year to 84 years old (median = 49.4 years), with the highest proportion in the 51–60 year age group (Figure 3). Cases were seen in each month, but the peak incidence occurred in August and September during the rainy season (from May to October) as seen in Figure 4. Of the 238 patients whose occupations were known, local farmers formed a large proportion (118, 49.6%). Among 277 cases with complete clinical information, septicaemia (153/277, 55.2%) and pneumonia (149/277, 53.8%) represented the two major clinical manifestations; other less common clinical features included musculo-skeletal or soft tissue abscesses (57/277, 20.6%), genitourinary or prostatic infection (18/277, 6.5%), brain infection (9/277, 3.3%), liver or splenic abscesses (8/277, 2.9%), pyogenic arthritis (5/277, 1.8%), upper gastrointestinal hemorrhage (4/277, 1.4%), neck abscess (4/277, 1.4%), suppurative parotitis (3/277, 1.1%), suppurative pharyngitis (3/277, 1.1%), and infected aortic aneurysm (3/277, 1.1%) as seen in Table 1. Death occurred in 64 cases, resulting in an overall mortality rate of 23.1% (64/277). The most common underlying co-morbidity was diabetes (131/277, 47.3%), followed by chronic liver disease (18/277, 6.5%), chronic lung disease (6/277, 2.2%), and chronic renal disease (5/277, 1.8%). During the survey period of 2011-2016, a notable peak in cases (*n* = 106) occurred in 2016, coinciding with record wet-season precipitation that year (Figure 2). This represented an annual incidence rate of 1.16 per 100,000 for this region, where the resident population was approximately 9,270,000 according to 2014 Census data. Even so, since this sentinel network did not include all the hospitals in the region and depended on voluntary surveillance rather than mandatory reporting, it is inevitable that a number of cases will have been missed and the true incidence would be higher.

To search for additional records of melioidosis cases and *B. pseudomallei* isolates from mainland China, especially those occurring before the start of the ICDC surveillance (2011), we also consulted the international *B. pseudomallei* multi-locus sequence typing (MLST) database (https://pubmlst.org/bpseudomallei/) [17]. Up to the end of 2017, among over 5300 *B. pseudomallei* isolates deposited in this web accessible database, 205 were marked as originating from the Chinese mainland, comprising 195 human/animal isolates, five environmental isolates, and five isolates with unknown sources. The isolation dates spanned from 1975 to 2015, but strains prior to 2011 accounted for only a small proportion (66 isolates). As there were many duplicate cases between this dataset and our surveillance database, we excluded these and have only included records prior to 2002 (11 isolates) in this review.

## 4. Literature Review

Alongside this increase in the number of cases diagnosed, there has been a proliferation of publications on melioidosis in China, many of which have been published in Chinese and are thus not readily accessible to researchers in other countries. In order to undertake as comprehensive a review of the available literature as possible, we conducted a search of two domestic science databases (China Science Periodical Database, China Hospital Knowledge Database) and two international scientific databases (PubMed of the National Library of Medicine and Web of Science) to identify reports of documented human or animal cases of melioidosis from China. We used the key search terms “melioidosis”, “*Burkholderia pseudomallei*” and “China”. We identified 196 articles to the end of August 2018, which we reviewed for relevance based on whether they described melioidosis cases or *B. pseudomallei* isolates from mainland China. A total of 86 articles (from 1981 to 2018) were finally selected, of which only 31 were retrieved from international databases. Most of these documents dealt with endemic provinces, but there were also case reports from non-endemic areas of central and northern China, including Beijing [18], Shanghai [19], Jiangsu [20,21], Hunan [22], Qinghai [23], Chongqing [24], Shaanxi [25], and Sichuan [26] as seen in Figure 2. Except for the Hunan case for whom the residence/travel information was unavailable, each case occurring outside the known endemic provinces of Hainan, Fujian, Guangdong, and Guangxi had a history of residence or traveled to Hainan or overseas endemic regions before the onset of the disease, hence all of them should be considered imported cases.

It is notable that several series of melioidosis cases have been described in China (Table 2) over the past two decades. However, in most series, the individual case data have been omitted or summarized, making it difficult to identify duplications as well as to investigate clinical or epidemiological patterns on a larger scale. In a retrospective study, Fang et al. described the clinical and epidemiological features of 170 culture-confirmed cases hospitalized in three general hospitals in Hainan, between 2002 and 2014 [35]. As noted above, a steady increase in the numbers of melioidosis cases was seen; pneumonia (34.1%) was the most common manifestation, and people with diabetes and outdoor laborers were at greatest risk of contracting the disease. In a prospective survey, Zheng et al. characterized a cluster of 16 microbiologically confirmed cases occurring after Typhoon Rammasun hit northern Hainan on July 18th 2014 [36]. In a retrospective study undertaken by Chen et al., a total of 44 human cases were detected amongst 7786 febrile patients at the Affiliated Hospital of Guangdong Medical College in Zhanjiang City between 1990 and 2005, 25 of whom died (56.8%) [28]. Since all these cases came from surrounding counties/villages, it appears that the Leizhou Peninsula, which has contributed to the majority of documented cases from Guangdong, should also be considered a hotspot for melioidosis in southern China. Compared with Hainan and Guangdong, fewer melioidosis cases have been recognized in the provinces of Guangxi and Fujian, despite the fact that *B. pseudomallei* is known to be present in the environment [9,41]. However, a retrospective report was recently published, which describes seven culture-proven cases of melioidosis hospitalized in a teaching hospital in Nanning, the capital city of Guangxi, from October 2006 to March 2015 [40]. The clinical characteristics, drug susceptibility results and epidemiological features of these cases were similar to those in other known endemic regions. In parallel, although not documented in the published literature, an indigenous female patient from Fujian with severe melioidosis was reported online in November 2014 [42]. Interestingly, the patient was from Putian, where the environmental presence of *B. pseudomallei* had previously been identified in 1985. It is thus likely that Guangxi, and possibly Fujian, have been neglected as melioidosis-endemic, and improvements in awareness and diagnosis of the disease are needed to improve the understanding of its true prevalence in these provinces. In general, the range of clinical manifestations of melioidosis in China is similar to that seen elsewhere, with relatively rare clinical manifestations reported in the literature, including suppurative parotitis (one case) [43], osteomyelitis (three cases) [44] and pericarditis (one case) [45] from Hainan, as well as prostatic abscesses (two cases) [46] and co-infection with Japanese encephalitis (one case) [47] from Guangdong.

In order to obtain a comprehensive dataset, we merged data from the three sources described above (ICDC, MLST database, and the literature review) to give an inclusive picture of the melioidosis distribution around mainland China (Appendix A), comprising 469 cases (401 from Hainan, 47 from Guangdong, 10 from Guangxi, one from Fujian and 10 cases imported into more northern provinces) identified in the present study. The distribution of these cases is shown in Figure 2.

## 5. Environmental and Molecular Investigations of *B. pseudomallei* in Hainan

### 5.1. Environmental Distribution of B. pseudomallei

The incidence of melioidosis is closely associated with the environmental density of *B. pseudomallei* in endemic areas. To estimate the abundance of the organism in soil and water and investigate its relationship with the recent increase in melioidosis cases, several environmental surveys have been conducted in Hainan over the past few years. In 2012, in a collaborative investigation by ICDC and Sanya People’s Hospital, 70 soil and water samples were collected from 20 paddy fields scattered across southern Hainan Island. There was no *B. pseudomallei* found, but four isolates of *Burkholderia thailandensis*, a non-pathogenic species closely related to *B. pseudomallei*, were recovered from four different locations [48]. In 2014 and 2017, two further investigations were conducted by separate research groups [49,50], which were positive for *B. pseudomallei* by culture in 3 of 58 (5.2%) and 2 of 70 (2.9%) environmental samples, respectively. Comparison of clinical and environmental isolates using a combination of two discriminant genotyping techniques, multilocus sequence typing and multiple locus variable number tandem repeat analysis (MLST and MLVA), suggested an environmental origin for some of the cases. These studies were of limited scale, and could not determine the geographic distribution of *B. pseudomallei* all over the island, but from August to December 2016, Dong et al. conducted a more comprehensive environmental *B. pseudomallei* investigation, covering all 18 counties of Hainan Island [51]. Among 360 sampling sites, 48 (13.3%) were positive for the bacterium, and most of these were located in coastal counties (12/18), which corresponds with the distribution of clinical cases around the island. It is hoped that studies such as this will help to establish a risk map, which may assist in the prevention of the disease in Hainan.

### 5.2. Genetic Diversity of B. pseudomallei Isolates from Hainan

Latterly, molecular epidemiological studies of the increasing numbers of available isolates of *B. pseudomallei* have been a growing area of melioidosis research in Hainan. Using MLST, Fang et al. divided 102 isolates from three hospitals into 41 sequence types (STs), among which 11 STs were unique to Hainan and eight were novel [52]. In another recent study, 30 STs were found amongst 60 clinical isolates collected between 2003 and 2014, including six novel types [16]. It is clear from these studies that *B. pseudomallei* in Hainan are highly genetically diverse, with some major STs being shared with and linked to the subpopulations from other Southeast Asian endemic foci such as Thailand and Malaysia. This makes it likely that the introduction of *B. pseudomallei* into Hainan is not recent and that the island has probably been endemic for melioidosis for a long period. Interestingly, ST562 has been found in both Hainan and northern Australia [53]. On the basis of SNP-based phylogenetic analysis, Price et al. showed that Australian isolates of this clone belong to the Southeast Asian subpopulation, which appears to have spread to, and become established in, the Darwin region in recent years [54]. ST562 is also represented amongst the two genomes (BPC006 and 350105) of *B. pseudomallei* from China that have been sequenced and published [55,56]. Strain 350105 was obtained from Hainan in 1976 and represents the earliest known ST562 isolate, but precisely how and when this long-range transmission took place is unknown and would be worthy of further studies.

## 6. New Understanding of Melioidosis Epidemiology on Hainan

### 6.1. The Association between Typhoon and Case Clustering of Melioidosis

Climatic factors can exert a remarkable influence on the incidence of melioidosis, even causing outbreaks. However, despite the known endemic areas of southern China (Hainan, Guangdong, Guangxi, Fujian, and Hong Kong) being located in the typhoon-prone belt of the Western Pacific, it was only recently discovered that the potential impact of a typhoon strike on melioidosis incidence in the region was demonstrated when Typhoon Rammasun (a Category five super typhoon) struck Hainan in 2014 [36]. Within two months of the event, a cluster of 16 human cases of melioidosis concurred around the point at which the typhoon struck land. Moreover, the patients manifested severe clinical forms and a high mortality rate (50%), which had also been observed in typhoon-related melioidosis clusters in Taiwan [57]. A correlation between typhoons and the occurrence of melioidosis was also noted in a review of 15 confirmed cases (2002–2007) from a municipal hospital in Zhanjiang City, Guangdong [58]. Further studies are required to elucidate the mode and mechanism by which extreme weather events influence the incidence of melioidosis in southern China and how this may be affected by climate change.

### 6.2. Disease Burden and Risk Analysis

Defining the burden and risk factors for melioidosis is important to raise awareness of the disease in the public health community and develop preventive and control strategies for the disease [4]. However, although an increasing number of melioidosis cases have been witnessed in Hainan, our knowledge of its full magnitude and epidemiology are still lacking. It appears that the main disease burden is focused on the tropical coastal areas of southern China, especially those of Hainan Island, the Leizhou Peninsula, and southern Guangxi. Compared with inland regions, these coastal regions are characterized by higher population density, more abundant rainfall and more plentiful habitats for *B. pseudomallei* (such as paddy fields and artificial ponds), which would favor the occurrence of melioidosis. As with other endemic regions, diabetic and rural populations are at the highest risk of contracting the disease, and the incidence is associated with rainfall and extreme weather events. The rising incidence of melioidosis has been more evident in Hainan than in the other endemic provinces in recent years. The epidemiology of melioidosis represents the consequences of a complicated interaction between multiple natural and social factors, such as the physicochemical environment, climate, host population susceptibility and genetic variation of the pathogen [3], which is as yet poorly understood. The impact of climate change (such as a rise in precipitation or higher frequency of typhoon strikes) on the incidence of disease may be significant in Hainan. In addition, the booming real estate industry on the island, which is likely to cause considerable changes in land use and the ecological environment, together with the rapid growth of the diabetic population in Hainan [59] may also be implicated in the recently observed increased incidence. Thus, multiple factors have probably contributed to a genuine increase in the incidence of melioidosis in Hainan, and unless these factors are addressed, it is likely that the background incidence will continue to rise in Hainan and, under appropriate conditions, further clusters and outbreaks may occur.

## 7. Surveillance and Reporting of Melioidosis Cases in Mainland China

### 7.1. Human

Although a highly efficient nationwide reporting and monitoring network for infectious diseases has been established in China since 2004 [60], melioidosis is still not included, largely due to the lack of awareness and the fact that it is not listed as a statutorily notifiable infectious disease in the country. Given the evidence of increasing incidence, ICDC has prioritized surveillance work in Hainan through the establishment of the sentinel network described above. Between 2011 and 2017, this network has involved seven tertiary and 12 county-level hospitals. In addition, in order to improve the ability of medical and laboratory staff to recognize the disease, ICDC delivered training and workshops on melioidosis diagnosis and *B. pseudomallei* identification during 2016 and 2017 in Hainan. However, no such surveillance or training has yet been undertaken in the Leizhou Peninsula and southern Guangxi, despite the evidence that these are also melioidosis-endemic. It is therefore likely that underreporting cases remains a major issue in these regions. Even though melioidosis has not yet been incorporated into the national system for reportable infectious diseases, the disease should be added into the local infectious disease reporting systems of these regions in the future.

Molecular epidemiological surveillance, which is widely accepted as a powerful tool for the prevention and control of communicable diseases, has increasingly been implemented in China. With the recognition of melioidosis as an emerging threat to both endemic and non-endemic areas, since 2017 *B. pseudomallei* has been included in PulseNet China, a national monitoring platform for major bacterial pathogens led by ICDC. This should enhance nationwide surveillance and enable studies of the molecular epidemiology of the disease across China.

### 7.2. Animals

Melioidosis can affect a wide range of animal species, but its true incidence in wild and domestic animals is far from clear in almost all parts of China. Following the initial identification of *B. pseudomallei* from environmental samples in Hainan, Lu et al. carried out surveys on farm animals and succeeded in recovering *B. pseudomallei* from the viscera of pigs and goats collected in the slaughterhouses of Haikou and Nanning in 1982 [10]. In 1993, a rhesus monkey was confirmed as having died of melioidosis in a health research centre in Fuzhou, Fujian province [61]. In 2005, Chen isolated *B. pseudomallei* from dead dolphins at a marine park in Sanya City, Hainan [62]. Additionally, a melioidosis outbreak was reported at a livestock farm in Guangxi in October 2009, resulting in the death of 12 Boer goats [63]. Based on these accounts, we believe that animal melioidosis probably affects a variety of animal species in southern China, although the true range and economic loss it causes are yet unknown.

### 7.3. Guidelines

No standards or official guidelines for the diagnosis or management of melioidosis are available in mainland China. Although serological tests such as IHA and ELISA were established and used to diagnose melioidosis in some hospitals in Hainan and Guangdong in the 1990s [8], in our experience, culture and the identification of *B. pseudomallei* have now been accepted as the consensus standard and is what has now been adopted for the diagnosis and reporting of the disease at hospitals and institutes in endemic regions of China.

## 8. Diagnosis and Treatment

Accurate and efficient diagnosis of melioidosis cases is crucial, both for saving lives and for understanding the epidemiology of the disease. Unfortunately, owing to the lack of a consensus guideline and the scarcity of studies, the true situation as far as melioidosis diagnosis is concerned is largely unknown in mainland China. However, what is known from the literature does not give cause for optimism [32,33,39]. Due to a lack of laboratory capacity and a lack of familiarity with the clinical characteristics of the disease amongst clinicians, melioidosis patients are likely to be misdiagnosed in most ordinary hospitals [32,33]; even in good tertiary hospitals, the rate of initial misdiagnosis is high (80%–90%) [39]. Given the difficulty of clinical diagnosis of this protean disease, culture and identification of *B. pseudomallei* is accepted as the gold standard for the diagnosis of melioidosis countrywide [14,32,33,35,39]. However, as yet this capacity is restricted to a few general or teaching hospitals within endemic regions of China, where automated blood culture and commercial bacterial identification systems are deployed and microbiological investigations are routinely conducted. In addition to the use of commercial identification systems (e.g., VITEK, BD Phoenix, etc.), identification of *B. pseudomallei* commonly depends on laboratory staff recognizing the basic morphological and cultural characteristics of *B. pseudomallei* [15,38]. However, waiting for the results of cultures may lead to delays in initiating effective treatment. In a retrospective survey of 40 melioidosis cases in Hainan, the time from admission to diagnosis was 4–19 days (average 8.2 days) [33]. To improve efficiency and accuracy, molecular technologies (e.g., specific PCR, 16s rRNA sequencing, etc.) have been applied for the confirmation of the diagnosis by specialist centers in recent years; however, no proven rapid screening approaches for *B. pseudomallei* (such as the latex agglutination assay [64] or even the ‘three disc test’ for the characteristic antimicrobial susceptibility pattern [65]) are available in most hospitals in endemic areas of China. However, computed tomography (CT) scans, magnetic resonance imaging (MRI), and ultrasound have increasingly been used to identify internal abscesses in the liver, spleen, prostate and brain [37,39]. Clearly, the capability for melioidosis diagnostics need to be increased, and efficient rapid screening methods for detecting *B. pseudomallei* need to be introduced to Hainan and other endemic regions of mainland China.

As in other melioidosis-endemic regions of the world, lung or liver abscesses caused by *B. pseudomallei* are often mistaken for tumors or tuberculosis in Hainan, as the clinical and imaging features are difficult to distinguish [66]. In addition, although extremely rare, a case of human glanders was documented in China in 2004 [67]. A fatal human case of human *B. thailandensis* infection was also reported in 2017 [68], although the validity of this has been questioned [69].

With the rise of the incidence of melioidosis, the major general hospitals in Hainan have gradually gained experience and capability in providing effective diagnosis and treatment of melioidosis. Once the diagnosis has been made, the treatment regimens used are consistent with international consensus guidelines and consist of an initial intensive phase with intravenous antibiotics and a subsequent eradication phase with oral antibiotics [40]. Compared with that of over 20 years ago, the overall mortality rate of acute melioidosis cases has lowered from 50%–66% [27,28] to 15%–25% [34,38,39]. Recurrence has been observed, but only in a tiny proportion of cases that have received standardized treatment [32,39]. Improvements in patient outcomes are probably due to a combination of factors: advances in diagnostic techniques, availability of appropriate drugs (meropenem, imipenem and ceftazidime) and the regular performance of antibiotic susceptibility tests during therapy [40]. Nevertheless, the differences between hospitals and areas in their ability to diagnose and treat patients with melioidosis, remains a major concern. Based on our observations, most remote county-level hospitals in Hainan lack both the diagnostic capacity and access to appropriate antimicrobial drugs. Furthermore, it is worth noting that the economic burden of treatment on poor rural patients and their families is significant, and some even choose to withdraw from treatment [14,31,32].

Additionally, although the susceptibility of *B. pseudomallei* to carbapenems is almost universal, the emergence of ceftazidime resistance has been reported as a problem in some hospitals in Hainan in more recent years. Ceftazidime resistance was said to have risen from 0% to 20–30% of *B. pseudomallei* clinical isolates between 2002 and 2014 in one institution in Hainan [70]. This requires confirmation, and warrants our continued surveillance and concern.

## 9. Current and Future Challenges

Over 40 years has passed since the first identification of *B. pseudomallei* in southern China; nonetheless, our knowledge on this disease is still lacking and there are numerous challenges to revealing its true epidemiology. In particular, some crucial questions remain unanswered, including the ecology and distribution of *B. pseudomallei* in the environment within China, the true burden of the disease in local residents, and its incidence among domestic and wild animals. Therefore, institutions, hospitals and laboratories in this region and countrywide need to collaborate closely with each other and undertake more comprehensive epidemiological and ecological studies to enhance our knowledge and understanding of the disease within China.

Moreover, the main endemic area is the coastal zone (Hainan, Guangdong, Guangxi and Fujian), which has experienced rapid development and has become one of the major economic centers of China. In view of its proximity to Southeast Asia, the world’s largest known melioidosis focus, and the booming economic and trade activities between them, close attention needs to be paid to the current and future trends of the disease in this area. In the meantime, under the national strategy of the Belt and Road Initiatives, Hainan has set the developmental goal of developing as a major international tourism island and seaborne trade centre in the next decade, with important progress towards these goals having already been made. In consequence, remarkable changes in the environment and population size can be anticipated for this region, which may increase the population at risk of infection with *B. pseudomallei* and pose greater challenges for the prevention and control of this disease in China and even regionally. Therefore, we recommend that melioidosis should be included in the national reportable diseases list, and that the sentinel surveillance network should be strengthened, which should lay a good foundation for risk assessment and the formulation of preventive strategies against this disease. Additionally, standard guidelines for the diagnosis and management of melioidosis need to be developed and distributed to all physicians and laboratory staff in both endemic and non-endemic areas of the country.

Although the national agency of disease control and prevention (ICDC) has set out to understand the current status through case and pathogen surveillance, the lack of awareness and attention of melioidosis amongst both government authorities and the general public is still a major challenge for the management and control of the disease in China. In a nationwide context, the public and the media hardly ever discuss this disease and threat it poses. Hence, we believe that a countrywide information campaign about melioidosis should be initiated in order to draw attention to this important disease.

## 10. Conclusions

In summary, this review presents a comprehensive account of the history, current status, and epidemiology of melioidosis in China. Surveillance data and a literature review clearly show that melioidosis has become more common in both endemic and non-endemic areas of mainland China, especially Hainan, in recent years, as in other parts of the world [71,72]. Surveillance and control strategies should be promoted to address this change, and further research should be conducted to provide new understanding and insights into the causes of this trend.

## Figures and Tables

**Figure 1 tropicalmed-04-00039-f001:**
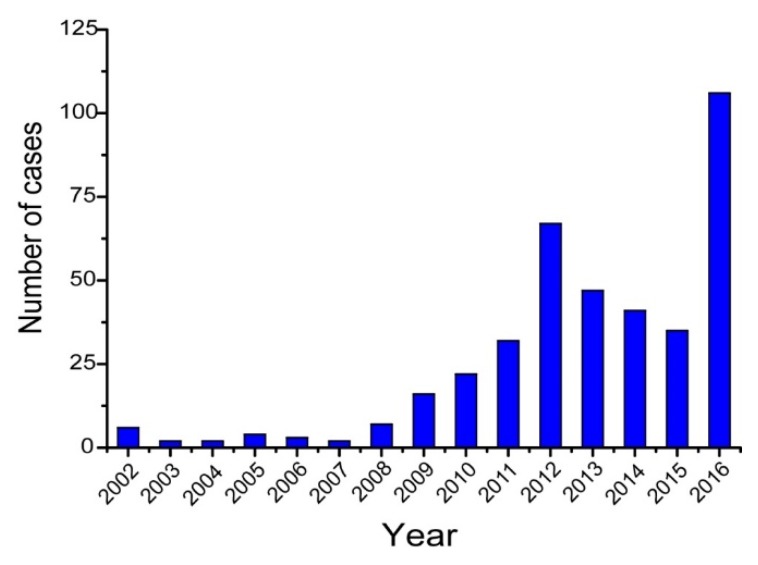
Number of melioidosis cases by year identified during surveillance, Hainan, China, 2002–2016.

**Figure 2 tropicalmed-04-00039-f002:**
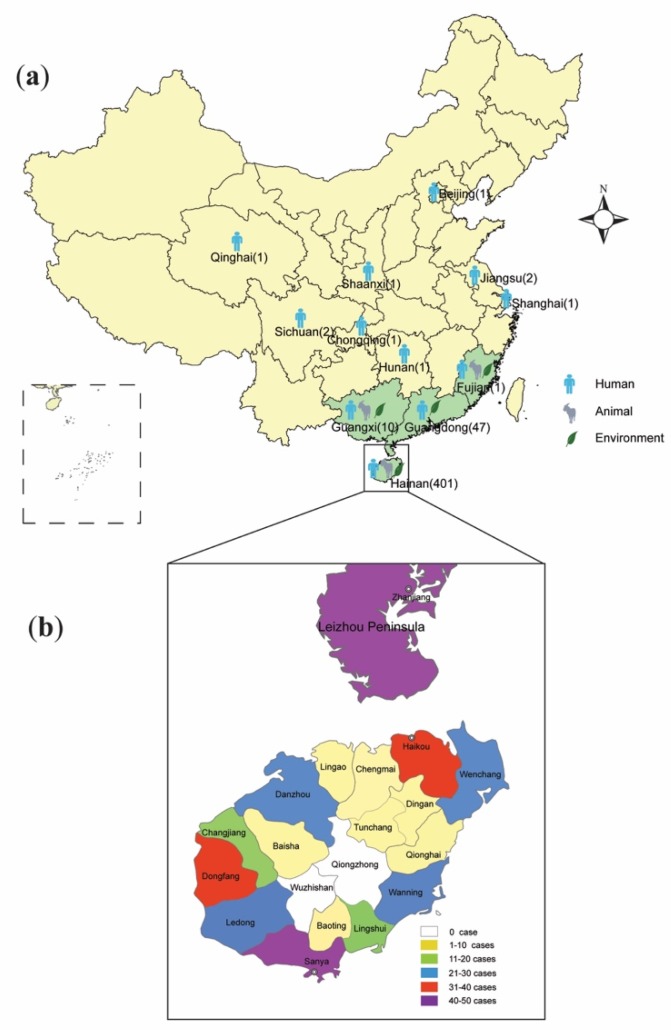
Location of *Burkholderia pseudomallei* and melioidosis cases in China. (**a**) The distribution of endemic areas (*n* = 4, with green shading) and melioidosis cases in mainland China, with locations of environmental isolation of *B. pseudomallei* and indigenous or imported human/animal cases of melioidosis indicated (The number of human cases for each province was indicated in parentheses. The patients outside the four endemic provinces all have a history of residence or travel to Hainan or overseas endemic areas and thus are deemed as imported cases); (**b**) The distribution of melioidosis cases according to different prefectures in Hainan (data only available for 289 of 401 cases) and in the Leizhou Peninsula (46 cases) of Guangdong, China.

**Figure 3 tropicalmed-04-00039-f003:**
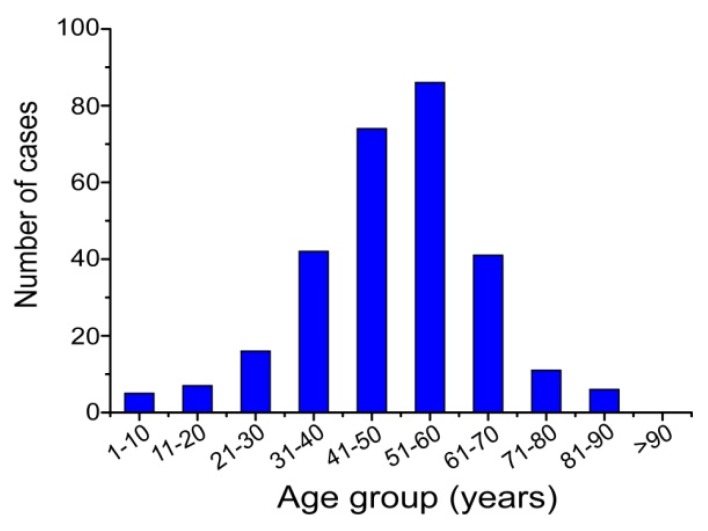
Age distribution of 288 human melioidosis patients from Hainan, China, 2002–2014.

**Figure 4 tropicalmed-04-00039-f004:**
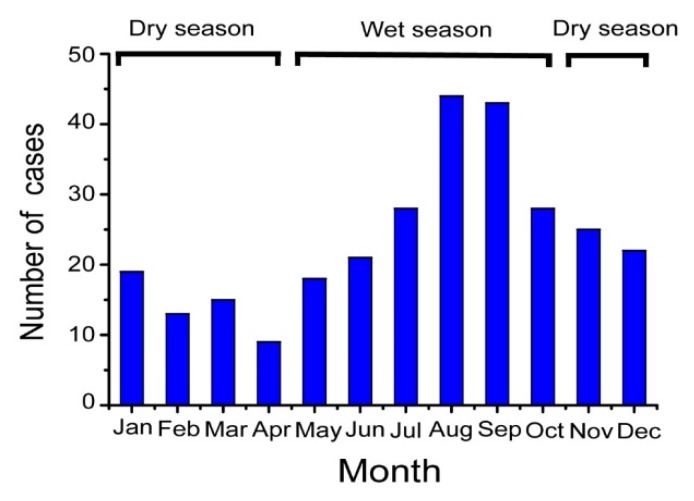
Monthly distribution of 289 melioidosis cases across 13 years (2002–2014) in Hainan, China.

**Table 1 tropicalmed-04-00039-t001:** Summary of clinical presentation of 277 melioidosis cases from Hainan.

Presentation	Number of Cases	Percentage (%)
Septicaemia	153	55.2
Pneumonia	149	53.8
Musculoskeletal or soft tissue abscess	57	20.6
Genitourinary or prostatic infection	18	6.5
Brain infection	9	3.3
Internal organ (liver, kidney or splenic) abscess	8	2.9
Pyogenic arthritis	5	1.8
Upper gastrointestinal hemorrhage	4	1.4
Neck abscess	4	1.4
Suppurative parotitis	3	1.1
Infected aortic aneurysm	3	1.1
Orbital abscess	3	1.1
Suppurative pharyngitis	3	1.1

**Table 2 tropicalmed-04-00039-t002:** Human melioidosis cases reported from Hainan, Guangdong, and Guangxi.

Year of Report	Location of Infection	Duration of Study	Number of Cases ^1^	Outcome	Reference
1992	Hainan & Guangdong	1975–1992	3	1 Survived/2 Died	[8]
1998	Hainan	1995–1996	8	4 Survived/2 Died/2 Unknown	[12]
2005	Hainan	2002–2005	12	4 Survived/8 Died	[27]
2006	Guangdong	1990–2005	44	19 Survived/25 Died	[28]
2006	Hainan	1996–2005	32	21 Survived/3 Died/9 treatment withdrawn	[29]
2008	Hainan	2002–2006	19	12 Survived/7 Died	[30]
2008	Hainan	2000–2007	25	9 Survived/6 Died/10 treatment withdrawn	[31]
2009	Hainan	2000–2009	104	72 Survived/15 Died/17 treatment withdrawn	[32]
2011	Hainan	2002–2008	122	81 Survived/19 Died/22 treatment withdrawn	[14]
2013	Hainan	2007–2012	95	65 Survived/30 Died	[15]
2014	Hainan	2009–2012	40	34 Survived/6 Died	[33]
2014	Hainan	2010–2013	40	6 Survived /10 Died/24 Unknown	[34]
2015	Hainan	2002–2014	170	124 Survived/46 Died	[35]
2016	Hainan	2003–2014	60	32 Survived/26 Died/2 Unknown	[16]
2016	Hainan	Jul–Sep 2014	16	8 Survived/8 Died	[36]
2017	Hainan	2002–2015	7	5 Survived (4 had nervous system sequelae) /1 Died/1 treatment withdrawn	[37]
2017	Hainan	2000–2012	46	33 Survived/13 Died	[38]
2018	Hainan	2012–2017	35	26 Survived/7 Died/2 treatment withdrawn	[39]
2018	Guangxi	2006–2015	7	All Survived	[40]

^1^ It was not possible to exclude duplicate reports and so there is overlap between these series.

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
