# Peer review of "Endemic Melioidosis in Southern China: Past and Present"

_tropicalmed, 2019, doi:10.3390/tropicalmed4010039_

Round 1
Reviewer 1 Report
I think overall the paper is a sold contribution to the literature. It does need more editing for English and grammar because there are multiple examples of awkward phrasing which leads to confusion by the reader. For example lines 29, 36-38, 46, 51, 54, 118-120 etc.
Also felt the paper could be shortened in every section (with careful editing) and convey the information in a more crisp and efficient manner.
Some comments
Lines 74-82 could be shortened considerably with a more objective tone
The environmental sampling section starting on line 247 deserves a separate section rather than lumping with human cases
New information section deserves a new title other than "new information"
Line 330 - no need for sentence limited knowledge is from literature
In diagnosis and treatment section starting on line 348 - I wish there were more references in this section instead of vague observations on diagnostics and general statements like lines 367 -372.
The resistance to ceftaz reported in a referenced paper is extremely important.
Again, sold contribution but needs another round of editing before re-submission in my opinion
Author Response
Reviewer 1:
COMMENT: I think overall the paper is a sold contribution to the literature. It does need more editing for English and grammar because there are multiple examples of awkward phrasing which leads to confusion by the reader. For example lines 29, 36-38, 46, 51, 54, 118-120 etc.
RESPONSE: Thank you for your comments and pointing out the grammar errors in the text. We agree with the criticism and have revised the incorrect wordings where possible so as to make the descriptions more readable.
COMMENT: Also felt the paper could be shortened in every section (with careful editing) and convey the information in a more crisp and efficient manner.
RESPONSE: We agree with the comment. We have substantially rewritten some sections and made careful editing to improve the English of the manuscript.
COMMENT: Lines 74-82 could be shortened considerably with a more objective tone
RESPONSE: We have revised this paragraph accordingly and now there is a more objective and concise description of this event. (Lines 61-66, page2)
COMMENT: The environmental sampling section starting on line 247 deserves a separate section rather than lumping with human cases
RESPONSE: We adopted the reviewer's suggestion. This paragraph has been reorganized and integrated it into the new separate section " Environmental and molecular investigations of B. pseudomallei in Hainan " .( Line 270, page7).
COMMENT: New information section deserves a new title other than "new information"
RESPONSE: The reviewer’s suggestions have been adopted and the title of this section has been changed to " New understanding on melioidosis epidemiology of Hainan ".(Line 272, page8)
COMMENT: Line 330 - no need for sentence limited knowledge is from literature
RESPONSE: This sentence has been eliminated.
COMMENT: In diagnosis and treatment section starting on line 348 - I wish there were more references in this section instead of vague observations on diagnostics and general statements like lines 367 -372.
RESPONSE: We agree with this criticism and have revised the manuscript accordingly. In the new edition of this part, we rewrote substantially and added more details and references to make the descriptions more readable and informative. (Lines 356-413, page10-11)
COMMENT: The resistance to ceftaz reported in a referenced paper is extremely important.
RESPONSE: No response required
COMMENT: Again, sold contribution but needs another round of editing before re-submission in my opinion
RESPONSE: We agree that the last version of the manuscript was rough and have edited the text carefully to improve the contents and English level of it in this new version.
Reviewer 2 Report
The authors aim at a risk analysis for melioidosis in China. They try to identify the areas of endemicity by screening literature especially Chinese publications which are often neglected if Western European or American reviewers are involved. For people in the field (public health and epidemiology) the data are interesting and deserve publication. The situation in tropical / subtropical China is explored comprehensively including human disease which is per se new for this country. The conclusions are consistent with what was written. The paper meets the standards of a good review. However, a comprehensive review and risk analysis for China is missing. The English language used is not easy to follow / understand, it needs editing so that the text is better readable. I cannot recommend to publish the text yet.
Author Response
Reviewer 2:
COMMENT: The authors aim at a risk analysis for melioidosis in China. They try to identify the areas of endemicity by screening literature especially Chinese publications which are often neglected if Western European or American reviewers are involved. For people in the field (public health and epidemiology) the data are interesting and deserve publication. The situation in tropical / subtropical China is explored comprehensively including human disease which is per se new for this country. The conclusions are consistent with what was written. The paper meets the standards of a good review. However, a comprehensive review and risk analysis for China is missing. The English language used is not easy to follow / understand, it needs editing so that the text is better readable. I cannot recommend to publish the text yet.
RESPONSE: We thank the reviewer for the comments. The suggestion on "Risk Analysis" has greatly inspired us. To address this issue, we have added a new part " Disease Burden and Risk Analysis " and integrated it into the new section " New understanding on melioidosis epidemiology of Hainan " of the manuscript.(Lines 287-312, page8-9)
We agree with the comment on the English level of the text. In this new version, we have substantially rewritten some sections and made every effort to improve the English of the manuscript.
Round 2
Reviewer 1 Report
substantial improvement in this version, thank you. Still needs some minor proofreading but an important contribution to the literature.
Author Response
Dear reviewer,
Thank you for your comments. The manuscropt has been further edited by the academic editor and the English has been greatly improved.
Best regrads,
Xiao Zheng